# Efficient Synthesis of 4,8-Dibromo Derivative of Strong Electron-Deficient Benzo[1,2-*d*:4,5-*d*’]bis([1,2,3]thiadiazole) and Its S_N_Ar and Cross-Coupling Reactions

**DOI:** 10.3390/molecules27217372

**Published:** 2022-10-30

**Authors:** Timofey N. Chmovzh, Daria A. Alekhina, Timofey A. Kudryashev, Oleg A. Rakitin

**Affiliations:** 1N. D. Zelinsky Institute of Organic Chemistry, Russian Academy of Sciences, 119991 Moscow, Russia; 2Nanotechnology Education and Research Center, South Ural State University, 454080 Chelyabinsk, Russia; 3Department of Chemistry, Moscow State University, 119899 Moscow, Russia

**Keywords:** sulfur–nitrogen heterocycles, benzo[1,2-*d*:4,5-*d*’]bis([1,2,3]thiadiazole), aromatic nucleophilic substitution, Suzuki and Stille cross-coupling reactions, X-ray analysis

## Abstract

An efficient synthesis of hydrolytically and thermally stable 4,8-dibromobenzo[1,2-*d*:4,5-*d*’]bis([1,2,3]thiadiazole) by the bromination of its parent heterocycle is reported. The structure of 4,8-dibromobenzo[1,2-*d*:4,5-*d*’]bis([1,2,3]thiadiazole) was confirmed by X-ray analysis. The conditions for the selective aromatic nucleophilic substitution of one bromine atom in this heterocyclic system by nitrogen nucleophiles are found, whereas thiols formed the bis-derivatives only. Suzuki–Miyaura cross-coupling reactions were found to be an effective method for the selective formation of various mono- and di(het)arylated derivatives of strong electron-deficient benzo[1,2-*d*:4,5-*d*’]bis([1,2,3]thiadiazole), and Stille coupling can be employed for the preparation of bis-arylated heterocycles, which can be considered as useful building blocks for the synthesis of DSSCs and OLEDs components.

## 1. Introduction

Organic photovoltaics is a developing technology with a unique set of properties, such as low-cost solution processing with nontoxic materials, the possibility of using small amounts of materials due to ultrathin absorber films, and the ease of varying the most important characteristics of materials [1]. The chromophores, both polymeric and low-molecular weight, used in these devices, consisting of a combination of electron-donating (D) and electron-withdrawing (A) groups linked either directly or, preferably, through a π-conjugated bridges (π), have been extensively studied. The electron-deficient π-conjugated building blocks play an essential role in achieving the most important characteristics, such as light absorption, light emission and charge carrier mobility in materials by reducing the bandgap by promoting intramolecular charge transfer (ICT) [2,3]. The most important property of the acceptor is the electron affinity, which is related to the energy of the lowest unoccupied molecular orbital (LUMO). The choice of acceptor is crucial to achieving high performance in materials such as bulk heterojunction solar cells (BHJ) [4,5], dye-sensitized solar cells (DSSCs) [4,6,7], n-type organic field-effect transistors [8,9], near-infrared absorption and emissions materials [10,11], and electrochromic materials [12,13]. Although many heterocyclic acceptor building blocks are known [14], 2,1,3-benzothiadiazole (BTD) occupies an exceptional place among them [15,16]. Nevertheless, there is a strong demand to create ultra-high electron-deficient building blocks based on it to improve the special characteristics of photovoltaic materials [17,18]. Several attempts have been made. The attachment of strong electron-withdrawing groups (i.e., fluorine) at positions 5 and 6 has produced many different materials derived from 5,6-difluoro-2,1,3-benzothiadiazole [13]. Recently, the synthesis of an ultra-high electron-deficient [1,2,5]thiadiazolo[3,4-*d*]pyridazine system with the formal replacement of carbon atoms in the 5- and/or 6-positions by electronegative atoms nitrogen has been reported [19]; it has been shown to be an important intermediate for the synthesis of dyes with various possible applications [18,20,21]. A third way to increase the electron-withdrawing strength of the benzothiadiazole ring is heteroannelation at positions 5 and 6 to create an acceptor, such as benzo-[1,2-*c*:4,5-*c*’]bis[1,2,5]thiadiazole (benzo-bis-thiadiazole, BBT), a sulfur–nitrogen heterocycle with the lowest LUMO energy [19]. The synthesis and functionalization of benzo[1,2-*c*:4,5-*c*’]bis[1,2,5]thiadiazole derivatives has been well investigated [22]. On the other hand, the possibilities of its bis(isothiadiazole) isomer, benzo[1,2-*d*:4,5-*d*’]bis([1,2,3]thiadiazole) (isoBTD), have been largely unexplored. Meanwhile, it has been shown that the replacement of the 1,2,5-thiadiazole ring in 2,1,3-benzothiadiazole (BTD) by the 1,2,3-thiadiazole ring leads to benzo[*d*][1,2,3]thiadiazole (isoBTD) compounds with properties similar to those of BTD, but with higher values of E_LUMO_ and the band gap (Eg), which indicates a high electronic conductivity due to the high stability of the molecule in the excited state. The careful selection of components in the design of appropriate compounds with benzo[*d*][1,2,3]thiadiazole as an internal acceptor can lead to promising photovoltaic materials [23]. In this regard, it becomes relevant to study the BBT isomer, benzo[1,2-*d*:4,5-*d*’]bis([1,2,3]thiadiazole) (isoBTD), which differs in the order of atoms in two five-membered heterocyclic rings. Figure 1 shows the boundary orbital energies (E_HOMO_ and E_LUMO_) for 2,1,3-benzothiadiazole, as well as its derivatives and isomers.

The data in Table 1 show that the fusion of a benzothiadiazole with another electron-withdrawing ring is the preferred method for increasing the electron-withdrawing capacity of the heterocyclic system, because it reduces the most important characteristic for photovoltaic materials, the energy band gap (E_g_). Nevertheless, E_g_ for isoBTD compounds is much higher than for BTD derivatives, which determines the stability of the molecule in the exited state, and as a result the high electron conductivity, and requires a careful selection of donor moieties in the dye by designing photovoltaic components.

Dihalogenated (usually dibrominated) derivatives of the electron-deficient heterocycles shown in Figure 1 are most often employed to prepare components of various photovoltaic materials [22,24,25]. 4,8-Dibromobenzo[1,2-*d*:4,5-*d*’]bis([1,2,3]thiadiazole) **1** was described as a minor component of an inseparable mixture of three compounds, with tricycle **2** and its mono-derivative **3** obtained by the nitrosation of 2,5-diaminobenzene-1,4-dithiol dihydrochoride **4** in HBr (Figure 1) [26]. Neither the yield of compound **1** nor its spectral and analytical data are given in the paper.

Herein, we describe the synthesis of 4,8-dibromobenzo[1,2-*d*:4,5-*d*’]bis([1,2,3]thiadiazole) **1**, its spectral and analytical data, S_N_Ar and cross-coupling reactions as a basis for the preparation of compounds that are of interest as photovoltaic materials.

## 2. Results and Discussion

### 2.1. Synthesis of 4,8-Dibromobenzo[1,2-d:4,5-d’]bis([1,2,3]thiadiazole) ***1***

Taking into account the failures in the synthesis of compound **1** from 2,5-diaminobenzene-1,4-dithiol dihydrochoride **4**, we decided to synthesize this compound from unsubstituted tricycle **2**, which is easily formed from dithiol **4** in almost quantitative yields upon nitrosation in hydrochloric acid [27]. We recently found that the bromination of tricycle **2** in hydrobromic acid can selectively give 4-bromobenzo[1,2-*d*:4,5-*d*’]bis([1,2,3]thiadiazole) **3** in good yields [28]. In a continuation of this study, the bromination of benzo[1,2-*d*:4,5-*d*’]bis([1,2,3]thiadiazole) **2** was carried out by varying the nature of the brominating agent (NBS and bromine), the solvent, and the reaction temperature. The results are summarized in Table 1.

The use of *N*-bromosuccinimide in a solvent (AcOH, CHCl_3_, DMF) or molecular bromine (Br_2_) in dioxane as brominating agents at room temperature did not lead to mono-**3** and di-bromides **1**; only starting compound **2** was isolated from the reaction mixture (Table 1, entries 1–6), which can be explained by the high electron-withdrawing character of tricycle **2**, which slows down the reaction of electrophilic bromination. Carrying out the reaction in hydrobromic acid with heating in the absence of molecular bromine also led to a low conversion of compound **2**, and 4-bromobenzo[1,2-*d*:4,5-*d*’]bis([1,2,3]thiadiazole) **3** was isolated in a 20% yield, while the yield of 4,8-dibromobenzo[1,2-*d*:4,5-*d*’]bis([1,2,3]thiadiazole) **1** did not exceed 5% (Entry 7). This result is similar to the formation of the mixture of compounds **1**, **2**, and **3,** which was obtained by the nitrosation of 2,5-diaminobenzene-1,4-dithiol dihydrochoride **4** in HBr (Figure 1). Carrying out the reaction when heated to 80 °C in hydrobromic acid in the presence of molecular bromine made it possible to increase the yield of 4-bromobenzo[1,2-*d*:4,5-*d*’]bis([1,2,3]thiadiazole) **3** to 60%. However, trace amounts of 4,8-dibromobenzo[1,2-*d*:4,5-*d*’]bis([1,2,3]thiadiazole) **1** were detected in the reaction mixture (Entry 8). Increasing both the reaction temperature to 120 °C and the amount of bromine made it possible to increase the yield of the target 4,8-dibromobenzo[1,2-*d*:4,5-*d*’]bis([1,2,3]thiadiazole) **1** up to 38%, while the yield of monobromo derivative **3** was only 5% (Entry 10). The longer heating of the reaction mixture (18 h) led to the target dibromide in a 40% yield (Entry 11). A further increase in the temperature of the reaction medium to 130 °C gave a slight decrease in the yield of the target compound (Entry 12). Thus, we have developed a method for the selective preparation of 4,8-dibromobenzo[1,2-*d*:4,5-*d*’]bis([1,2,3]thiadiazole) **1** in moderate yield. Unfortunately, the reactivity of the parent ultra-high electron-withdrawing heterocycles, such as benzo-[1,2-*c*:4,5-*c*’]bis[1,2,5]thiadiazole (BBT) or benzo[1,2-*d*:4,5-*d*’]bis([1,2,3]thiadiazole) **2,** has not been previously studied. We assume that for such compounds, the reactions of the electrophilic substitution of the hydrogen atom are difficult, and it is necessary to apply the conditions of radical reactions (for example, bromine in hydrobromic acid). The reactivity of the unsubstituted tricycle **2** will be reported in more detail in our next publication.

4,8-Dibromobenzo[1,2-*d*:4,5-*d*’]bis([1,2,3]thiadiazole) **1** is a high-melting yellow solid with mp > 250 °C (Appendix A). It is stable to hydrolysis at room temperature and can be kept in a freezer under argon for a few months without noticeable changes. Its structure was fully proven by NMR, mass and IR spectroscopy, and finally established by single-crystal X-ray diffraction study (Figure 2).

### 2.2. Aromatic Nucleophilic Substitution of 4,8-Dibromobenzo[1,2-d:4,5-d’]bis([1,2,3]thiadiazole) ***1***

The nucleophilic substitution in 1,2,5-thiadiazoles and 1,2,3-thiadiazoles fused with benzene or strong electron-withdrawing heterocycles has been poorly investigated. Two examples of nucleophilic substitution with morpholine and piperidine are known for 4,7-dibromobenzo[*c*][1,2,5]thiadiazole **5** [29,30], one reaction with morpholine for 4,7-dibromobenzo[*d*][1,2,3]thiadiazole **6** [31] and two procedures with morpholine and thiophenol for benzo[1,2-*c*:4,5-*c*’]bis[1,2,5]thiadiazole **7** [32]. The formation of mono-substituted derivatives **8,9** in the first two cases and bis-derivative **10** for the more activated tricycle **7** was observed (Figure 2).

Aromatic nucleophilic substitution in 4,7-dibromo[1,2,5]thiadiazolo[3,4-*d*]pyridazine **11** was recently investigated [14], and selective conditions were found for the substitution of one or two bromine atoms by oxygen and nitrogen nucleophiles, whereas thiols formed only bis-derivatives (Figure 3).

In order to find selective conditions for the formation of mono- and bis-substituted products in 4,8-dibromobenzo[1,2-*d*:4,5-*d*’]bis([1,2,3]thiadiazole) **1**, its reactions with nitrogen, sulfur- and oxygen-containing nucleophiles were systematically studied.

The treatment of 4,8-dibromobenzo[1,2-*d*:4,5-*d*’]bis([1,2,3]thiadiazole) **1** with morpholine gave mono- and bis-substituted derivatives, depending on the reaction conditions. Carrying out the reaction at room temperature, regardless of the solvent used (DCM, MeCN, DMF) and the amount of morpholine, resulted in the mono-substitution product **12a** in low to moderate yields (Table 2, entries 1–6). Refluxing of the reaction mixture in MeCN for 24 h with two morpholine equivalents gave a mono-substituted product **12a** with the best yield of 78% (Entry 7). Heating in DMF under similar conditions did not improve the yield of **12a** (70%, entry 8). The formation of bis-substituted product **13a** in a moderate yield was achieved by the prolonged heating of dibromide **1**, with an excess of morpholine at 130 °C (Entry 9). Thus, optimal conditions were found for the synthesis of an asymmetric compound **12a** and a disubstituted product **13a**.

We applied the conditions found for nucleophilic substitution reactions to other primary and secondary amines. Piperidine **14b** was found to react similarly to morpholine, forming mono-**12b** and bis-**13b** derivatives with moderate yields (Table 3, entries 1 and 2). A somewhat unexpected result was obtained for pyrrolidine **14c**, which was converted into a mono-derivative **12c** both when carrying out the reaction in acetonitrile and when heated at 130 °C in DMF; in the second case, the yield of the mono-product **12c** decreased, apparently due to its partial decomposition at elevated temperatures (Entries 3,4). With the cyclopentaindoline derivative **14d** and aniline **14e**, dibromide **1** reacted when heated to 100–130 °C in DMF with an excess of amine, to form only mono-substitution products **12d**,**e** in moderate yields (Entries 6–9). However, with more basic aliphatic primary amines, such as cyclohexylamine **14f** and *tert*-butylamine **14g**, no nucleophilic aromatic substitution reaction products were isolated, and only the partial decomposition of the starting dibromide **1** occurred when heated to 100–130 °C in DMF (Entries 10–16). When mono-adduct **12a** and cyclohexylamine were heated in DMF at 130 °C, the latter decomposed within 12 h to form a mixture of products, from which it was not possible to isolate identifiable compounds.

It was shown that carbazole and diphenylamine did not react with 4,8-dibromobenzo[1,2-*d*:4,5-*d*’]bis([1,2,3]thiadiazole) **1** under the studied conditions. Our attempts to carry out reactions using sodium salts of these amines (obtained in situ from carbazole or diphenylamine and NaH) were also unsuccessful—dibromide **1** decomposed when reacted with sodium carbazolate or sodium diphenylamide in THF or DMF for 3 h, and heated at a temperature of 60 °C.

Thus, we have successfully developed effective methods for introducing aromatic and aliphatic amines into the 4,8-dibromobenzo[1,2-*d*:4,5-*d*’]bis([1,2,3]thiadiazole) molecule **1** with the selective formation of mono-**12** and bis-**13** substitution products (Appendix A). Nucleophilic aromatic substitution reactions under mild conditions led to the formation of the mono-substitution product **12**. The introduction of a donor fragment into the dibromide molecule leads to a decrease in the reactivity of position 4 or 7 of the benzene ring. Therefore, to replace the second bromine atom, it is necessary to apply severe conditions, namely, to heat the reaction mixture in DMF to 130 °C.

The reactions of 4,8-dibromobenzo[1,2-*d*:4,5-*d*’]bis([1,2,3]thiadiazole) **1** with such *S*-nucleophiles as thiophenol, hexylthiol, and dodecanethiol were studied. It was shown that when dibromide **1** was treated with thiophenol at room temperature in THF or DMF, the formation of only trace amounts of dimercapto derivative **15a** was observed, even when using less than a two-fold equivalent of thiol (Table 4, entries 1,2). The use of one equivalent of base and thiophenol resulted in a mixture of disubstituted derivative **15a** and starting dibromide **1**, and our attempts to isolate the mono-substituted derivative were unsuccessful. The reaction with two equivalents of sodium thiophenolate led to an increase in the yield of bis(phenylthio) derivative **15a** to 80% (Entry 3). The optimal conditions for the preparation of **15a** were extended to hexanethiol and dodecanothiol, which resulted in the preparation of bis-thiols **15b**,**c** in high yields (Appendix A).

Dibromo derivative **1** proved to be sufficiently resistant to hydrolysis. Our study of the reaction of dibromide with various *O*-nucleophiles, such as alcohols (water, methanol, ethanol, *n*-butanol, *tert*-butanol, phenol) and corresponding alcoholates, showed that 4,8-dibromobenzo[1,2-*d*:4,5-*d*’]bis([1,2,3]thiadiazole) **1** did not react with them in both THF and DMF. The heating of the reaction mixtures also did not lead to the nucleophilic substitution of bromine atoms, and starting compound **1** was isolated in all cases.

### 2.3. Cross-Coupling Reactions of 4,8-Dibromobenzo[1,2-d:4,5-d’]bis([1,2,3]thiadiazole) ***1***

4,8-Dibromobenzo[1,2-*d*:4,5-*d*’]bis([1,2,3]thiadiazole) **1** was investigated in palladium-catalyzed Suzuki–Miyaura and Stille coupling reactions in order to obtain mono- and bis-aryl(hetaryl)thiadiazolopyridazines. The mono-arylation of dibromo derivatives of strong electron-deficient heterocycles is the most challenging task, and requires special attention to achieve the best yields [15,19,25].

#### 2.3.1. Suzuki–Miyaura Coupling

The behavior of 4,8-dibromobenzo[1,2-*d*:4,5-*d*’]bis([1,2,3]thiadiazole) **1** in the palladium-catalyzed Suzuki–Miyaura coupling reactions was investigated in order to find the best conditions for the preparation of mono- and bis-aryl derivatives. In the reaction of dibromo derivative **1** with thiophen-2-yl-pinacolate ester **16a** and boronic acid **17a**, the base, solvent, and reaction temperature were varied. The results of this study are summarized in Table 5. We have shown that the nature of the reagents, solvents, and the temperature of the reaction medium significantly affect the course of chemical transformations. The tetrakis(triphenylphosphine)palladium complex (Pd(PPh_3_)_4_) most widely used in these transformations was employed as a catalyst. It was shown that when the reaction with pinacolate ester **16a** (1 eqv) was carried out at 110 °C in toluene for 24 h, the formation of the mono-coupling product **18a** was observed in 72% yield (Table 6, entry 1). The presence of water to dissolve inorganic salts reduced the yield of product **18a** to 50%, which is apparently associated to side reactions (Entry 2). We have shown that when using 2-thiopheneboronic acid **17a**, the mono-coupling product **18a** was isolated in almost the same yield (70%) as thiophen-2-pinacolate ester **16a** (cf. entries 1, 3). Since the use of 2-thiopheneboronic acid **17a** did not lead to an increase in the yield of the target product **18a**, and also because to the lower stability of 2-thiopheneboronic acids **17**, pinacolate esters of boronic acids **16** were employed in further studies. Carrying out the reaction in xylene at 130 °C with two eqv. of pinacolate ester **16a** resulted in the bis-coupling product **19a** with a 65% yield (Entry 4). The best conditions for the synthesis of mono-**18a** and bis-**19a** products were extended to other pinacolate esters **16b**–**h**. In all cases, mono-coupling products **18b**–**h** and bis-products **19b**–**h** were selectively formed by refluxing the reaction mixtures in toluene with one eqv. **16** (**18b**–**h**) or in xylene with two eqv. **17** (**19b**–**h**) in moderate yields (50–72%) (Entries 5–18).

Thus, it was found that the Suzuki reaction between 4,8-dibromobenzo[1,2-*d*:4,5-*d*’]bis([1,2,3]thiadiazole) **1** and pinacolate esters **16** requires the use of anhydrous solvents, and mono-**18** and bis-**19** coupling products can be selectively obtained by changing the reaction temperature (Appendix A).

#### 2.3.2. Stille Coupling

We studied the Stille reaction of 4,8-dibromobenzo[1,2-*d*:4,5-*d*’]bis([1,2,3]thiadiazole) **1** with various aromatic and heteroaromatic stannyl derivatives **20a**–**h** to synthesize both mono- and bis-coupling products. Optimal conditions were developed for the reaction with thienyltributyl stannate **20a** in toluene in the presence of PdCl_2_(PPh_3_)_2_ as the most widely used catalyst in these transformations. When refluxing in toluene, the reaction could not be stopped at the stage of the mono-aryl derivative’s **18a** formation, even when one equivalent of stannate **20a** was used (Table 6, entry 1). By employing two equivalents of stannate **20a**, the bis-coupling product **19a** was isolated in high yields (Entry 2). Careful temperature control allowed us to find the best conditions for the synthesis of the mono-aryl derivative **18a** by heating the reaction mixture at 60 °C for 16 h, although we could not avoid the formation of the bis-product in small amounts (Entry 3). An increase in the reaction temperature to 80 °C led to a decrease in the selectivity of the formation of the mono-coupling product **18a**, and a decrease in the reaction to 40 °C led to a decrease in the yield of the product **18a** (Entries 4,5). The conditions found for the formation of bis-**19a** and mono-**18a** adducts (Entries 2,3) were applied to other aryl(hetaryl) stannates **20**. As a result, a series of mono- **18a**–**h** and bis-substitution products **19a**–**h** were isolated in moderate to high yields (Entries 6–19).

Thus, we have shown that the Stille reactions, compared with the Suzuki reactions, proceeded faster and with similar yields, but less selectively in the case of the formation of mono-coupling products **18**.

## 3. Experimental Section

### 3.1. Materials and Reagents

The chemicals were purchased from commercial sources (Sigma-Aldrich, St. Louis, MO, USA) and used as received. Benzo[1,2-*d*:4,5-*d*’]bis([1,2,3]thiadiazole) **2** [27], 4,4,5,5-tetramethyl-2-(thiophen-2-yl)-1,3,2-dioxaborolane **16a** [33], 2-(4-hexylthiophen-2-yl)-4,4,5,5-tetramethyl-1,3,2-dioxaborolane **16b** [34], 2-([2,2′-bithiophen]-5-yl)-4,4,5,5-tetramethyl-1,3,2-dioxaborolane **16c** [35], (4-(diphenylamino)phenyl)boronic acid **17h** [36], tributyl(4-hexyl-2-thienyl)stannane **20b** [37], [2,2′-bithiophen]-5-yltributylstannane **20c** [38], tributyl(5-(2-ethylhexyl)thiophen-2-yl)stannane **20d** [39], tributyl(*p*-tolyl)stannane **20f** [40], tributyl(4-methoxyphenyl)stannane **20g** [40], and diphenyl-(4-tributylstannanyl-phenyl)-amine **20h** [41] were prepared according to the published methods and characterized by NMR spectra. All synthetic operations were performed under a dry argon atmosphere. The solvents were purified by distillation over the appropriate drying agents.

### 3.2. Analytical Instruments

The melting points were determined on a Kofler hot-stage apparatus and were uncorrected. ^1^H and ^13^C NMR spectra were taken with a Bruker AM-300 machine (Bruker Ltd., Moscow, Russia) with TMS as the standard. *J* values are given in Hz. MS spectra (EI, 70 eV) were obtained with a Finnigan MAT INCOS 50 instrument (Thermo Finnigan LLC, San Jose, CA, USA). High-resolution MS spectra were measured on a Bruker micrOTOF II instrument using electrospray ionization (ESI). IR spectra were measured with a Bruker “Alpha-T” instrument (Bruker, Billerica, MA, USA) in KBr pellets. The HOMO–LUMO energies were calculated using the Gaussian 16 Rev C.01 program M11 DFT functional with a 6–31+g(d) basis.

### 3.3. X-ray Analysis

X-ray diffraction data were collected at 100 K on a four-circle Rigaku Synergy S diffractometer equipped with a HyPix600HE area-detector (kappa geometry, shutterless ω-scan technique), using graphite mono-chromatized Cu K_α_-radiation. The intensity data were integrated and corrected for absorption and decay by the CrysAlisPro program (Austin, TX, USA, accessed on 1 September 2022) [42]. The structure was solved by direct methods using SHELXT and refined on *F*^2^ using SHELXL-2018 [43] in the OLEX2 program [44]. All non-hydrogen atoms were refined with individual anisotropic displacement parameters. All hydrogen atoms were placed in ideal calculated positions and refined as riding atoms with relative isotropic displacement parameters. A rotating group model was applied for methyl groups. The Cambridge Crystallographic Data Centre holds the supplementary crystallographic data for this paper (Table 7), No. CCDC 2210625 (compound **1**). These data can be obtained free of charge via http://www.ccdc.cam.ac.uk/conts/retrieving.html, accessed on 1 September 2022 (or from the CCDC, 12 Union Road, Cambridge CB2 1EZ, UK; Fax: +44-1223-336033; E-mail: deposit@ccdc.cam.ac.uk).

### 3.4. 4,8-Dibromobenzo[1,2-d:4,5-d’]bis([1,2,3]thiadiazole) ***1***

Bromine (2.5 mL, 48.55 mmol) was added dropwise to a solution of compound **2** (650 mg, 3.35 mmol) in HBr (18 mL) at 80 °C. The reaction mixture was stirred for 18 h at 120 °C, cooled to room temperature, poured onto ice, washed with NaHSO_3_, extracted with EtOAc and dried over MgSO_4_. The solvent was evaporated under reduced pressure. The residue was purified by column chromatography (silica gel Merck 60, CH_2_Cl_2_) to give the title compound **1**. Yield 471 mg (40%), yellow solid. Eluent—CH_2_Cl_2_/hexane, 1:2 (*v*/*v*). Mp > 250 °C. (R_f_ = 0.3, CH_2_Cl_2_). IR *ν*_max_ (KBr, cm^–1^): 2925, 2853, 1300, 1287, 1191, 1160, 1082, 968, 890, 811, 547, 511. ^13^C NMR (75 MHz, CDCl_3_): *δ* 153.3, 143.4, 105.0. HRMS (ESI-MS), *m*/*z*: calcd for C_6_^79^Br_2_N_4_S_2_ [M + Ag]^+^, 456.6977, found, 456.6977. MS, *m*/*z* (%): 353 ([M + 1]^+^, 60), 352 (M^+^, 80), 351 ([M − 1]^+^, 58), 296 (100), 217 (62), 173 (70), 149 (85), 136 (68), 92 (86), 68 (75), 44 (10), 28 (5).

### 3.5. General Procedure for the Preparation of Mono-Aminated Products ***12a***–***c***

Amine **14a**–**c** (0.29 mmol) was added to a solution of 4,8-dibromobenzo[1,2-*d*:4,5-*d*’]bis([1,2,3]thiadiazole) **1** (50 mg, 0.14 mmol) in dry MeCN (10 mL) at room temperature and the mixture was stirred at reflux for 18 h, poured into water (20 mL) and extracted with CH_2_Cl_2_ (3 × 35 mL). The combined organic layers were washed with brine, dried over MgSO_4_, filtered, and concentrated under reduced pressure. The crude product was purified by column chromatography (silica gel Merck 60).

#### 3.5.1. 4-(8-Bromobenzo[1,2-*d*:4,5-*d*’]bis([1,2,3]thiadiazole)-4-yl)morpholine (**12a**)

Orange solid, 39 mg (78%), eluent—CH_2_Cl_2_/hexane, 2:1 (*v*/*v*). R_f_ = 0.1 (CH_2_Cl_2_:hexane, 1:1, (*v*/*v*)). Mp = 150–153 °C. IR *ν*_max_ (KBr, cm^–1^): 2960, 2920, 2857, 1640, 1531, 1458, 1432, 1376, 1341, 1313, 1277, 1249, 1187, 1113, 1029, 984, 874, 815, 631, 511. ^1^H NMR (300 MHz, CDCl_3_): *δ* 4.04–4.01 (m, 4H), 3.99–3.95 (m, 4H). ^13^C NMR (75 MHz, CDCl_3_): *δ* 157.5, 148.4, 145.9, 138.9, 131.2, 94.7, 67.4, 52.3. HRMS (ESI-TOF), *m*/*z*: calcd for C_10_H_9_^79^BrN_5_OS_2_ [M + H]^+^, 357.9426, found, 357.9417. MS (EI, 70 eV), *m*/*z* (*I*, %): 359 ([M + 1]^+^, 12), 358 ([M]^+^, 11), 331 (14), 245 (30), 149 (31), 43 (35), 28 (100).

#### 3.5.2. 4-Bromo-8-(piperidin-1-yl)benzo[1,2-*d*:4,5-*d*’]bis([1,2,3]thiadiazole) (**12b**)

Orange solid, 37 mg (75%), eluent—CH_2_Cl_2_/hexane, 1:1 (*v*/*v*). R_f_ = 0.4 (CH_2_Cl_2_:hexane, 1:1, (*v*/*v*)). Mp = 143–145 °C. IR *ν*_max_ (KBr, cm^–1^): 2936, 2846, 1641, 1531, 1442, 1376, 1344, 1315, 1264, 1238, 1188, 1153, 1099, 1019, 979, 890, 865, 847, 817, 603, 539. ^1^H NMR (300 MHz, CDCl_3_): *δ* 3.97–3.94 (m, 4H), 1.94–1.84 (m, 6H). ^13^C NMR (75 MHz, CDCl_3_): *δ* 157.5, 148.1, 145.9, 140.0, 131.0, 92.57, 53.5, 26.8, 24.4. HRMS (ESI-TOF), *m*/*z*: calcd for C_11_H_10_^79^BrN_5_S_2_ [M]^+^, 354.9556, found, 354.9563. MS (EI, 70 eV), *m*/*z* (*I*, %): 357 ([M + 1]^+^, 5), 356 ([M]^+^, 4), 329 (27), 317 (24), 212 (80), 105 (78), 77 (45), 55 (40), 41 (100), 28 (78).

#### 3.5.3. 4-Bromo-8-(pyrrolidin-1-yl)benzo[1,2-*d*:4,5-*d*’]bis([1,2,3]thiadiazole) (**12c**)

Orange solid, 36 mg (77%), eluent—CH_2_Cl_2_/hexane, 1:2 (*v*/*v*). R_f_ = 0.3 (CH_2_Cl_2_:hexane, 1:1, (*v*/*v*)). Mp = 153–155 °C. IR *ν*_max_ (KBr, cm^–1^): 2955, 2921, 1854, 1638, 1540, 1447, 1373, 1354, 1323, 1303, 1260, 1236, 1177, 1119, 991, 893, 870, 816, 739, 667, 515. ^1^H NMR (300 MHz, CDCl_3_): *δ* 4.30 (t, *J* = 6.5, 4H), 2.23–2.19 (m, 4H). ^13^C NMR (75 MHz, CDCl_3_): *δ* 157.6, 146.6, 144.7, 138.3, 127.1, 87.0, 53.8, 29.8. HRMS (ESI-TOF), *m*/*z*: calcd for C_10_H_9_^79^BrN_5_S_2_ [M + H]^+^, 341.9477, found, 341.9475. MS (EI, 70 eV), *m*/*z* (*I*, %): 344 ([M + 2]^+^, 3), 343 ([M + 1]^+^, 50), 342 (M^+^, 4), 341 ([M − 1]^+^, 46), 315 (55), 285 (25), 256 (23), 206 (40), 191 (35), 178 (37), 149 (86), 96 (48), 69 (50), 57 (100), 41 (95), 27 (47), 18 (84).

### 3.6. General Procedure for the Preparation of Mono-Aminated Products ***12d***,***e***

Aniline **14d**, **14e** (27 mg, 0.29 mmol) was added to a solution of 4,8-dibromobenzo[1,2-*d*:4,5-*d*’]bis([1,2,3]thiadiazole) **2** (50 mg, 0.14 mmol) in dry DMF (10 mL), and the mixture was stirred at 100–130 °C for 18 h, poured into water and extracted with CH_2_Cl_2_ (3 × 35 mL). The combined organic layers were washed with water and brine, dried over MgSO_4_, filtered, and concentrated under reduced pressure. The crude product was purified by column chromatography (silica gel Merck 60).

#### 3.6.1. 4-Bromo-8-(2,3,3*a*,8*b*-tetrahydrocyclopenta[*b*]indol-4(1*H*)-yl)benzo[1,2-*d*:4,5-*d*’]bis([1,2,3]thiadiazole) (**12d**)

Violet solid, yield 36 mg (60%), eluent—CH_2_Cl_2_/hexane, 1:2 (*v*/*v*)). R_f_ = 0.4 (CH_2_Cl_2_/hexane, 1:1 (*v*/*v*)). Mp = 178–180 °C. IR *ν*_max_ (KBr, cm^–1^): 2957, 2924, 2853, 1597, 1523, 1480, 1444, 1366, 1306, 1259, 1187, 1080, 1023, 968, 881, 813, 749, 695, 535, 523. ^1^H NMR (300 MHz, CDCl_3_): *δ* 7.32–7.26 (m, 2H), 7.11 (t, *J* = 7.7, 1H), 6.99 (t, *J* = 7.4, 1H), 6.36–6.30 (m, 1H), 4.03–3.97 (m, 1H), 2.12–1.94 (m, 2H), 1.74–1.45 (m, 4H). ^13^C NMR (75 MHz, CDCl_3_): *δ* 157.7, 150.5, 145.3, 144.5, 136.4, 126.8, 125.6, 124.5, 124.1, 122.2, 111.0, 97.6, 72.8, 46.8, 35.2, 33.7, 24.8. HRMS (ESI-TOF), *m*/*z*: calcd for C_17_H_12_^79^BrN_5_S_2_ [M]^+^, 428.9712, found, 428.9708. MS (EI, 70 eV), *m*/*z* (*I*, %): 432 ([M + 2]^+^, 4), 431 ([M + 1]^+^, 39), 430 (M^+^, 3), 429 ([M − 1]^+^, 41), 403 (58), 375 (26), 346 (53), 147 (30), 115 (28), 67 (50), 57 (47), 41 (100), 27 (80).

#### 3.6.2. 8-Bromo-*N*-phenylbenzo[1,2-*d*:4,5-*d*’]bis([1,2,3]thiadiazole)-4-amine (**12e**)

Red solid, yield 50 mg (55%), eluent—CH_2_Cl_2_/hexane, 1:1 (*v*/*v*)). R_f_ = 0.3 (CH_2_Cl_2_/hexane, 1:1 (*v*/*v*)). Mp = 157–160 °C. IR *ν*_max_ (KBr, cm^–1^): 2958, 2924, 2853, 1729, 1548, 1495, 1465, 1450, 1425, 1323, 1287, 1262, 1099, 1078, 1022, 967, 868, 810, 743, 717, 696, 533. ^1^H NMR (300 MHz, CDCl_3_): *δ* 8.28 (s, 1H), 7.56–7.50 (m, 3H), 7.32 (dd, *J* = 7.5, 1.5, 2H). ^13^C NMR (75 MHz, CDCl_3_): *δ* 158.3, 144.8, 144.4, 136.6, 134.1, 129.8, 128.2, 127.0, 123.4, 91.3. HRMS (ESI-TOF), *m*/*z*: calcd for C_12_H_7_^79^BrN_5_S_2_ [M + H]^+^, 363.9321, found, 363.9321. MS (EI, 70 eV), *m*/*z* (*I*, %): 364 ([M]^+^, 4), 363 ([M − 1]^+^, 3), 337 (20), 228 (40), 184 (45), 149 (20), 125 (35), 77 (98), 51 (100), 28 (23).

### 3.7. General Procedure for the Preparation of Bis-Aminated Products ***13a***,***b***

Amine **14** (0.84 mmol) was added to a solution of 4,8-dibromobenzo[1,2-*d*:4,5-*d*’]bis([1,2,3]thiadiazole) **1** (50 mg, 0.14 mmol) in dry DMF (10 mL) at room temperature and the mixture was stirred at 130 °C for 24 h, poured into water (20 mL) and extracted with CH_2_Cl_2_ (3 × 35 mL). The combined organic layers were washed with brine, dried over MgSO_4_, filtered, and concentrated under reduced pressure. The crude product was purified by column chromatography (silica gel Merck 60).

#### 3.7.1. 4,8-Dimorpholinobenzo[1,2-*d*:4,5-*d*’]bis([1,2,3]thiadiazole) (**13a**)

Red solid, 20 mg (40%), eluent—CH_2_Cl_2_. R_f_ = 0.1 (CH_2_Cl_2_). Mp > 250 °C. IR *ν*_max_ (KBr, cm^–1^): 2968, 2947, 2918, 2849, 1827, 1637, 1539, 1480, 1442, 1371, 1327, 1298, 1272, 1251, 1108, 1068, 992, 922, 887, 824, 632, 514. ^1^H NMR (300 MHz, CDCl_3_): *δ* 4.01–3.98 (m, 8H), 3.77–3.74 (m, 8H). ^13^C NMR (75 MHz, CDCl_3_): *δ* 153.0, 135.7, 133.2, 67.7, 52.6. HRMS (ESI-TOF), *m*/*z*: calcd for C_14_H_16_N_6_O_2_S_2_ [M]^+^, 364.0771, found, 364.0769. MS (EI, 70 eV), *m*/*z* (*I*, %): 364 ([M]^+^, 12), 192 (20), 45 (21), 28 (100).

#### 3.7.2. 4,8-Di(piperidin-1-yl)benzo[1,2-*d*:4,5-*d*’]bis([1,2,3]thiadiazole) (**13b**)

Red solid, 20 mg (50%), eluent—CH_2_Cl_2_/hexane, 1:2 (*v*/*v*). R_f_ = 0.6 (CH_2_Cl_2_:hexane, 1:1 (*v*/*v*)). Mp = 158–160 °C. IR *ν*_max_ (KBr, cm^–1^): 2928, 2847, 2804, 1637, 1479, 1446, 1372, 1327, 1313, 1268, 1241, 1202, 1112, 1072, 1056, 1030, 982, 855, 823, 737, 701, 614, 515. ^1^H NMR (300 MHz, CDCl_3_): *δ* 3.68–3.64 (m, 8H), 1.87–1.73 (m, 12H). ^13^C NMR (75 MHz, CDCl_3_): *δ* 153.0, 135.8, 134.0, 53.7, 27.0, 24.5. HRMS (ESI-TOF), *m*/*z*: calcd for C_16_H_20_N_6_S_2_ [M]^+^, 360.1185, found, 360.1183. MS (EI, 70 eV), *m*/*z* (*I*, %): 360 ([M]^+^, 20), 303 (21), 275 (14), 247 (19), 219 (22), 192 (15), 96 (35), 55 (40), 41 (100), 29 (20).

### 3.8. General Procedure for the Reaction of 4,8-Dibromobenzo[1,2-d:4,5-d’]Bis([1,2,3]Thiadiazole) ***1*** with Thiols

Sodium hydride (6 mg, 0.28 mmol) was added to a solution of thiol (0.28 mmol) in dry THF (15 mL) at 0 °C with stirring. The reaction mixture was stirred at 0 °C for 30 min, then 4,8-dibromobenzo[1,2-*d*:4,5-*d*’]bis([1,2,3]thiadiazole) **1** (50 mg, 0.14 mmol) was added. The mixture was stirred for 6 h at room temperature. On completion (monitored by TLC), the mixture was poured into water (20 mL) and extracted with CH_2_Cl_2_ (3 × 5 mL). The combined organic layers were washed with brine, dried over MgSO_4_, filtered, and concentrated under reduced pressure. The crude product was purified by column chromatography.

#### 3.8.1. 4,8-Bis(phenylthio)benzo[1,2-*d*:4,5-*d*’]bis([1,2,3]thiadiazole) (**15a**)

Orange solid, 45 mg (80%), Mp = 168–172 °C, eluent—CH_2_Cl_2_/hexane, 1:2 (*v*/*v*). R_f_ = 0.4 (CH_2_Cl_2_/hexane, 1:1 (*v*/*v*)). IR *ν*_max_ (KBr, cm^–1^): 1574, 1492, 1468, 1438, 1397, 1371, 1310, 1265, 1231, 1155, 1065, 1020, 1000, 907, 815, 748, 688, 566, 500. ^1^H NMR (300 MHz, CDCl_3_): δ 7.66–7.58 (m, 4H), 7.54–7.40 (m, 6H). ^13^C NMR (75 MHz, CDCl_3_): δ 156.3, 141.4, 135.7, 135.0, 130.2, 130.0, 123.8. HRMS (ESI-TOF), *m*/*z*: calcd for C_18_H_11_N_4_S_4_ [M + H]^+^, 410.9861, found, 410.9849. MS (EI, 70 eV), *m*/*z* (*I*, %): 410 ([M]^+^, 100), 320 (80), 290 (35), 277 (37), 201 (15), 177 (16), 169 (13), 136 (13), 77 (50), 51 (15).

#### 3.8.2. 4,8-Bis(hexylthio)benzo[1,2-*d*:4,5-*d*’]bis([1,2,3]thiadiazole) (**15b**)

Yellow solid, 46 mg (78%), eluent—CH_2_Cl_2_/hexane, 1:4 (*v*/*v*). R_f_ = 0.7 (CH_2_Cl_2_/hexane, 1:1 (*v*/*v*)). Mp = 78–80 °C. IR *ν*_max_ (KBr, cm^–1^): 2957, 2924, 2853, 1637, 1461, 1412, 1375, 1315, 1280, 1239, 1179, 906, 811, 723, 515. ^1^H NMR (300 MHz, CDCl_3_): δ 3.62 (t, *J* = 7.4, 4H), 1.66 (p, *J* = 7.3, 4H), 1.48–1.39 (m, 4H), 1.27–1.21 (m, 8H), 0.84 (t, *J* = 7.0, 6H). ^13^C NMR (75 MHz, CDCl_3_): δ 155.7, 145.3, 122.5, 36.9, 31.2, 30.0, 28.2, 22.4, 13.9. HRMS (ESI-TOF), *m*/*z*: calcd for C_18_H_27_N_4_S_4_ [M + H]^+^, 427.1113, found, 427.1108. MS (EI, 70 eV), *m*/*z* (*I*, %): 426 ([M]^+^, 4), 314 (M^+^, 3), 215 (4), 136 (5), 101 (7), 55 (25), 43 (100), 29 (95).

#### 3.8.3. 4,8-Bis(dodecylthio)benzo[1,2-*d*:4,5-*d*’]bis([1,2,3]thiadiazole) (**15c**)

Yellow solid, 46 mg (78%), eluent—CH_2_Cl_2_/hexane, 1:2 (*v*/*v*). R_f_ = 0.9 (CH_2_Cl_2_). Mp = 78–80 °C. IR *ν*_max_ (KBr, cm^–1^): 2955, 2922, 2851, 1642, 1469, 1414, 1375, 1312, 1262, 1238, 1177, 1084, 1026, 903, 806, 719, 517. ^1^H NMR (300 MHz, CDCl_3_): δ 3.62 (t, *J* = 7.3, 4H), 1.65 (p, *J* = 7.3, 4H), 1.47–1.38 (m, 4H), 1.32–1.21 (m, 32H), 0.87 (t, *J* = 6.6, 6H). ^13^C NMR (75 MHz, CDCl_3_): δ 155.7, 145.3, 122.5, 36.9, 31.9, 30.0, 29.7, 29.6, 29.58, 29.51, 29.4, 29.0, 28.5, 22.7, 14.1. HRMS (ESI-TOF), *m*/*z*: calcd for C_30_H_50_N_4_S_4_Ag [M]^+^, 701.1964, found, 701.1954. MS (EI, 70 eV), *m*/*z* (*I*, %): 595 ([M]^+^, 12), 215 (8), 97 (3), 83 (5), 69 (18), 57 (50), 43 (100), 29 (35).

### 3.9. General Procedure for the Preparation of Mono-Substituted Products ***18*** under Suzuki Coupling Conditions (Procedure A)

A mixture of 4,8-dibromobenzo[1,2-*d*:4,5-*d*’]bis([1,2,3]thiadiazole) **1** (50 mg, 0.14 mmol), boronic ether **16a**–**h** or its acid **17a** (0.14 mmol), K_2_CO_3_ (19 mg, 0.14 mmol), and Pd(PPh_3_)_4_ (24 mg, 15% mmol) in dry toluene (8 mL) was degassed by argon and heated at 110 °C in a sealed vial. On completion (monitored by TLC), the mixture was poured into water and extracted with CH_2_Cl_2_ (3 × 35 mL). The combined organic layers were washed with brine, dried over MgSO_4_, filtered, and concentrated under reduced pressure. The crude product was purified by column chromatography.

### 3.10. General Procedure for the Preparation of Bis-Substituted Products ***19*** under Suzuki Coupling Conditions (Procedure B)

A mixture of 4,8-dibromobenzo[1,2-*d*:4,5-*d*’]bis([1,2,3]thiadiazole) **1** (50 mg, 0.14 mmol), boronic ether **16a**–**h** or its acid **17a** (0.28 mmol), K_2_CO_3_ (38 mg, 0.28 mmol), and Pd(PPh_3_)_4_ (24 mg, 15% mmol) in dry xylene (8 mL) was degassed by argon and heated at 130 °C in a sealed vial. On completion (monitored by TLC), the mixture was poured into water and extracted with CH_2_Cl_2_ (3 × 35 mL). The combined organic layers were washed with brine, dried over MgSO_4_, filtered, and concentrated under reduced pressure. The crude product was purified by column chromatography.

### 3.11. General Procedure for the Preparation of Mono-Substituted Products ***18*** under Stille Coupling Conditions (Procedure C)

PdCl_2_(PPh_3_)_2_ (14 mg, 15% mmol) and stannane **20a**–**h** (0.14 mmol) were added to a solution of 4,8-dibromobenzo[1,2-*d*:4,5-*d*’]bis([1,2,3]thiadiazole) **1** (50 mg, 0.14 mmol) in anhydrous toluene (4 mL) The resulting cloudy yellow mixture was stirred and degassed by argon in a sealed vial. The resulting yellow mixture was then stirred at 60 °C for the desired time. On completion (monitored by TLC), the mixture was washed with water and the organic layer was extracted with CH_2_Cl_2_ (3 × 35 mL), dried over MgSO_4_ and then concentrated in vacuo. The products were isolated by column chromatography.

### 3.12. General Procedure for the Preparation of Bis-Substituted Products ***19*** under Stille Coupling Conditions (Procedure D)

PdCl_2_(PPh_3_)_2_ (14 mg, 15% mmol) and stannane **20a**–**h** (0.28 mmol) were added to a solution of 4,8-dibromobenzo[1,2-*d*:4,5-*d*’]bis([1,2,3]thiadiazole) **1** (50 mg, 0.14 mmol) in anhydrous toluene (4 mL) The resulting cloudy yellow mixture was stirred and degassed by argon in a sealed vial. The resulting yellow mixture was then stirred at 110 °C for the desired time. On completion (monitored by TLC), the mixture was washed with water and the organic layer was extracted with CH_2_Cl_2_ (3 × 35 mL), dried over MgSO_4_ and then concentrated in vacuo. The products were isolated by column chromatography.

#### 3.12.1. 4-Bromo-8-(thiophen-2-yl)benzo[1,2-*d*:4,5-*d*’]bis([1,2,3]thiadiazole) (**18a**)

Yellow solid, 35 mg (72%, procedure A) or 34 mg (70%, procedure C), eluent—CH_2_Cl_2_:hexane, 1:1 (*v*/*v*).R_f_ = 0.4 (CH_2_Cl_2_). Mp = 198–200 °C. IR *ν*_max_ (KBr, cm^–1^): 1738, 1641, 1494, 1464, 1413, 1262, 1186, 1081, 1023, 809, 701, 544. ^1^H NMR (300 MHz, CDCl_3_): *δ* 8.20 (d, *J* = 3.9, 1H), 7.76 (d, *J* = 5.2, 1H), 7.36 (t, *J* = 4.5, 1H). ^13^C NMR (75 MHz, CDCl_3_): *δ* 156.1, 152.2, 139.32, 139.1, 137.0, 131.7, 130.8, 128.8, 122.4, 104.4. HRMS (ESI-TOF), *m*/*z*: calcd for C_10_H_4_^79^BrN_4_S_3_ [M + H]^+^, 354.8776, found, 354.8772. MS (EI, 70 eV), *m*/*z* (*I*, %): 356 ([M + 2]^+^, 2), 355 ([M + 1]^+^, 31), 354 ([M]^+^, 35), 353 ([M − 1]^+^, 6), 328 (100), 298 (10), 247 (40), 219 (35), 175 (42), 151 (75), 117 (13), 93 (32), 69 (20), 45 (22).

#### 3.12.2. 4-Bromo-8-(4-hexylthiophen-2-yl)benzo[1,2-*d*:4,5-*d*’]bis([1,2,3]thiadiazole) (**18b**)

Yellow solid, 43 mg (70%, procedure A) or 44 mg (73%, procedure C), eluent—CH_2_Cl_2_:hexane, 1:2 (*v*/*v*). R_f_ = 0.6 (CH_2_Cl_2_:hexane, 1:1). Mp = 67–69 °C. IR *ν*_max_ (KBr, cm^–1^): 2954, 2924, 2852, 1642, 1449, 1397, 1371, 1327, 1306, 1263, 1237, 1109, 1089, 892, 812, 727, 581, 537. ^1^H NMR (300 MHz, CDCl_3_): *δ* 8.09 (s, 1H), 7.35 (s, 1H), 2.76 (t, *J* = 7.7, 2H), 1.73 (p, *J* = 7.2, 2H), 1.42–1.31 (m, 6H), 0.91 (t, *J* = 6.9, 3H). ^13^C NMR (75 MHz, CDCl_3_): *δ* 155.9, 152.0, 145.4, 144.7, 138.8, 136.4, 133.3, 125.6, 122.5, 103.8, 31.6, 30.5, 30.4, 28.9, 22.6, 14.1. HRMS (ESI-TOF), *m*/*z*: calcd for C_16_H_16_^79^BrN_4_S_3_ [M + H]^+^, 438.9715, found, 438.9730. MS (EI, 70 eV), *m*/*z* (*I*, %): 440 ([M + 1]^+^, 3), 439 ([M]^+^, 2), 412 (3), 355 (2), 312 (2), 235 (7), 164 (8), 125 (11), 111 (15), 97 (50), 83 (52), 69 (60), 57 (98), 43 (100), 29 (75).

#### 3.12.3. 4-([2,2′-Bithiophen]-5-yl)-8-bromobenzo[1,2-*d*:4,5-*d*’]bis([1,2,3]thiadiazole) (**18c**)

Red solid, 39 mg (65%, procedure A) or 41 mg (67%, procedure C), eluent—CH_2_Cl_2_:hexane, 1:1 (*v*/*v*). R_f_ = 0.4 (CH_2_Cl_2_:hexane, 1:1 (*v*/*v*)). Mp = 130–132 °C. IR *ν*_max_ (KBr, cm^–1^): 1639, 1445, 1423, 1375, 1308, 1268, 1123, 892, 812, 689, 537. ^1^H NMR (300 MHz, CDCl_3_): *δ* 8.09 (d, *J* = 4.0, 1H), 7.40–7.34 (m, 3H), 7.14–7.07 (m, 1H). ^13^C NMR (75 MHz, CDCl_3_): *δ* 156.3, 151.9, 144.9, 143.5, 138.4, 136.2, 135.2, 132.6, 128.4, 126.3, 125.4, 125.0, 122.0, 103.9. HRMS (ESI-TOF), *m*/*z*: calcd for C_14_H_6_^79^BrN_4_S_4_ [M + H]^+^, 436.8653, found, 436.8642. MS (EI, 70 eV), *m*/*z* (*I*, %): 439 ([M + 2]^+^, 5), 438 ([M]^+^, 43), 437 ([M]^+^, 4), 436 ([M − 1], 39), 410 (37), 382 (10), 329 (52), 301 (20), 257 (62), 233 (71), 225 (20), 149 (60), 127 (55), 117 (53), 93 (70), 69 (100), 45 (95).

#### 3.12.4. 4-Bromo-8-(5-(2-ethylhexyl)thiophen-2-yl)benzo[1,2-*d*:4,5-*d*’]bis([1,2,3]thiadiazole) (**18d**)

Yellow solid, 44 mg (68%, procedure A) or 45 mg (69%, procedure C), eluent—CH_2_Cl_2_:hexane, 1:1 (*v*/*v*). R_f_ = 0.6 (CH_2_Cl_2_). Mp = 57–60 °C. IR *ν*_max_ (KBr, cm^–1^): 2955, 2922, 2853, 1639, 1455, 1421, 1376, 1307, 1266, 1213, 1104, 892, 813, 544. ^1^H NMR (300 MHz, CDCl_3_): *δ* 8.06 (d, *J* = 3.8, 1H), 7.01 (d, *J* = 3.8, 1H), 2.90 (d, *J* = 6.8, 2H), 1.77–1.69 (m, 1H), 1.45–1.31 (m, 8H), 0.97–0.89 (m, 6H). ^13^C NMR (75 MHz, CDCl_3_): *δ* 156.0, 151.8, 151.6, 144.7, 138.3, 134.4, 131.8, 126.9, 122.7, 103.1, 41.5, 34.4, 32.4, 29.7, 25.6, 23.0, 14.1, 10.8. HRMS (ESI-TOF), *m*/*z*: calcd for C_18_H_20_^79^BrN_4_S_3_ [M + H]^+^, 467.0028, found, 467.0019. MS (EI, 70 eV), *m*/*z* (*I*, %): 470 ([M + 2]^+^, 1), 469 ([M + 1]^+^, 2), 468 ([M]^+^, 11), 467 ([M − 1]^+^, 2), 466 ([M − 2]^+^, 9), 440 (6), 369 (3), 341 (5), 326 (4), 232 (5), 188 (10), 105 (30), 83 (35), 71 (60), 57 (100), 43 (95), 29 (15).

#### 3.12.5. 4-Bromo-8-phenylbenzo[1,2-*d*:4,5-*d*’]bis([1,2,3]thiadiazole) (**18e**)

Yellow solid, 34 mg (70%, procedure A) or 32 mg (67%, procedure C), eluent—CH_2_Cl_2_:hexane, 1:1 (*v*/*v*). R_f_ = 0.4 (CH_2_Cl_2_). Mp = 163–165 °C. IR *ν*_max_ (KBr, cm^–1^): 1725, 1445, 1398, 1371, 1343, 1316, 1275, 1211, 1188, 1155, 1078, 1025, 924, 898, 823, 766, 698, 538. ^1^H NMR (300 MHz, CDCl_3_): *δ* 7.97–7.93 (m, 2H), 7.68–7.61 (m, 3H). ^13^C NMR (75 MHz, CDCl_3_): *δ* 155.8, 153.9, 144.6, 141.6, 136.3, 130.5, 129.5, 129.4, 128.9, 105.3. HRMS (ESI-TOF), *m*/*z*: calcd for C_12_H_6_^79^BrN_4_S_2_ [M + H]^+^, 348.9212, found, 348.9222. MS (EI, 70 eV), *m*/*z* (*I*, %): 350 ([M+]^+^, 2), 349 ([M]^+^, 1), 322 (20), 241 (50), 213 (18), 169 (96), 145 (100), 137 (15), 117 (18), 93 (35), 69 (45), 43 (30).

#### 3.12.6. 4-Bromo-8-(*p*-tolyl)benzo[1,2-*d*:4,5-*d*’]bis([1,2,3]thiadiazole) (**18f**)

Yellow solid, 34 mg (67%, procedure A) or 35 mg (68%, procedure C), eluent—CH_2_Cl_2_:hexane, 1:1 (*v*/*v*). R_f_ = 0.5 (CH_2_Cl_2_). Mp = 135–137 °C. IR *ν*_max_ (KBr, cm^–1^): 2958, 2924, 2854, 1729, 1493, 1461, 1401, 1385, 1283, 1187, 1081, 1023, 969, 894, 812, 723, 491. ^1^H NMR (300 MHz, CDCl_3_): *δ* 7.90 (d, *J* = 7.9, 2H), 7.46 (d, *J* = 7.9, 2H), 2.52 (s, 3H). ^13^C NMR (75 MHz, CDCl_3_): *δ* 155.8, 153.8, 147.2, 144.6, 141.0, 133.4, 130.2, 129.4, 128.1, 104.8, 21.6. HRMS (ESI-TOF), *m*/*z*: calcd for C_13_H_8_^79^BrN_4_S_2_ [M + H]^+^, 362.9368, found, 362.9370. MS (EI, 70 eV), *m*/*z* (*I*, %): 363 ([M]^+^, 3), 362 ([M − 1]^+^, 1), 361 ([M − 2]^+^, 5), 360 ([M − 3]^+^, 5), 359 ([M − 4]^+^, 2), 318 (98), 303 (15), 285 (16), 159 (98), 115 (97), 93 (50), 69 (85), 57 (100), 43 (96).

#### 3.12.7. 4-Bromo-8-(4-methoxyphenyl)benzo[1,2-*d*:4,5-*d*’]bis([1,2,3]thiadiazole) (**18g**)

Yellow solid, 32 mg (60%, procedure A) or 26 mg (50%, procedure C), eluent—CH_2_Cl_2_:hexane, 1:1 (*v*/*v*). R_f_ = 0.2 (CH_2_Cl_2_). Mp = 125–127 °C. IR *ν*_max_ (KBr, cm^–1^): 2957, 2925, 2854, 1641, 1609, 1502, 1460, 1429, 1376, 1301, 1264, 1179, 1081, 1029, 967, 885, 829, 815, 702, 614, 534. ^1^H NMR (300 MHz, CDCl_3_): *δ* 7.92 (d, *J* = 8.7, 2H), 7.16 (d, *J* = 8.7, 2H), 3.94 (s, 3H). ^13^C NMR (75 MHz, CDCl_3_): *δ* 161.3, 155.8, 153.8, 144.6, 141.3, 131.1, 128.9, 128.5, 114.9, 104.3, 55.6. HRMS (ESI-TOF), *m*/*z*: calcd for C_13_H_8_^79^BrN_4_OS_2_ [M + H]^+^, 378.9317, found, 378.9314. MS (EI, 70 eV), *m*/*z* (*I*, %): 380 ([M + 1]^+^, 7), 379 ([M]^+^, 5), 350 (20), 271 (80), 199 (90), 175 (100), 132 (55), 28 (9).

#### 3.12.8. 4-(8-Bromobenzo[1,2-*d*:4,5-*d*’]bis([1,2,3]thiadiazole)-4-yl)-*N*,*N*-diphenylaniline (**18h**)

Red solid, 52 mg (72%, procedure A) or 46 mg (64%, procedure C), eluent—CH_2_Cl_2_:hexane, 1:2 (*v*/*v*). R_f_ = 0.5 (CH_2_Cl_2_:hexane, 1:1 (*v*/*v*)). Mp = 165–168 °C. IR *ν*_max_ (KBr, cm^–1^): 1588, 1468, 1432, 1331, 1315, 1274, 1193, 1075, 1026, 814, 753, 695, 618, 512. ^1^H NMR (300 MHz, CDCl_3_): *δ* 7.83 (d, *J* = 8.7, 2H), 7.35 (t, *J* = 7.7, 4H), 7.24–7.08 (m, 8H). ^13^C NMR (75 MHz, CDCl_3_): *δ* 155.9, 153.7, 150.0, 146.8, 144.7, 141.0, 130.6, 129.7, 129.1, 128.4, 125.9, 124.5, 121.4, 103.8. HRMS (ESI-TOF), *m*/*z*: calcd for C_24_H_14_^79^BrN_5_S_2_ [M]^+^, 514.9869, found, 514.9874. MS (EI, 70 eV), *m*/*z* (*I*, %): 519 ([M + 2]^+^, 5), 518 ([M + 1]^+^, 10), 517 ([M]^+^, 75), 516 ([M − 1]^+^, 11), 515 ([M − 2]^+^, 70), 489 (8), 459 (6), 408 (50), 380 (35), 336 (100), 312 (65), 167 (12), 149 (15), 77 (20), 51 (11).

#### 3.12.9. 4,8-Di(thiophen-2-yl)benzo[1,2-*d*:4,5-*d*’]bis([1,2,3]thiadiazole) (**19a**)

Red solid, 32 mg (65%, procedure B) or 33 mg (67%, procedure D), eluent—CH_2_Cl_2_:hexane, 1:2 (*v*/*v*). R_f_ = 0.5 (CH_2_Cl_2_:hexane, 1:1 (*v*/*v*)). Mp > 250 °C. IR *ν*_max_ (KBr, cm^–1^): 1637, 1513, 1442, 1412, 1376, 1341, 1315, 1267, 1064, 819, 797, 700, 549. ^1^H NMR (300 MHz, CDCl_3_): *δ* 8.20 (dd, *J* = 3.8, 1.2, 2H), 7.74 (dd, *J* = 5.1, 1.2, 2H), 7.37 (dd, *J* = 5.1, 3.8, 2H). ^13^C NMR (75 MHz, CDCl_3_): *δ* 154.1, 139.6, 137.7, 131.1, 130.2, 128.4, 120.2. HRMS (ESI-TOF), *m*/*z*: calcd for C_14_H_7_N_4_S_4_ [M + H]^+^, 358.9548, found, 358.9544. MS (EI, 70 eV), *m*/*z* (*I*, %): 360 ([M + 2]^+^, 21), 359 ([M + 1]^+^, 22), 358 ([M]^+^, 96), 330 (45), 302 (98), 270 (30), 226 (28), 151 (50), 69 (48), 57 (85), 43 (100), 29 (99), 17 (98).

#### 3.12.10. 4,8-Bis(4-hexylthiophen-2-yl)benzo[1,2-*d*:4,5-*d*’]bis([1,2,3]thiadiazole) (**19b**)

Red solid, 46 mg (63%, procedure B) or 47 mg (57%, procedure D), eluent—CH_2_Cl_2_:hexane, 1:3 (*v*/*v*). R_f_ = 0.7 (CH_2_Cl_2_:hexane, 1:1 (*v*/*v*)). Mp = 134–136 °C. IR *ν*_max_ (KBr, cm^–1^): 2961, 2928, 2851, 1645, 1461, 1180, 1071, 925, 761, 555, 459. ^1^H NMR (300 MHz, CDCl_3_): *δ* 8.09 (d, *J* = 1.3, 2H), 7.32 (d, *J* = 1.3, 2H), 2.77 (t, *J* = 7.7, 4H), 1.75 (p, *J* = 7.6, 4H), 1.45–1.33 (m, 12H), 0.94–0.89 (m, 6H). ^13^C NMR (75 MHz, CDCl_3_): *δ* 154.0, 145.0, 139.4, 137.4, 132.7, 125.0, 120.8, 31.6, 30.5, 30.4, 29.0, 22.6, 14.1. HRMS (ESI-TOF), *m*/*z*: calcd for C_26_H_31_N_4_S_4_ [M + H]^+^, 527.1426, found, 527.1407. MS (EI, 70 eV), *m*/*z* (*I*, %): 528 ([M + 2]^+^, 37), 527 ([M + 1]^+^, 50), 526 ([M]^+^, 98), 479 (11), 455 (40), 441 (93), 427 (30), 413 (42), 329 (25), 235 (40), 165 (98), 120 (70), 105 (99), 69 (96), 55 (99), 29 (100).

#### 3.12.11. 4,8-Di([2,2′-bithiophen]-5-yl)benzo[1,2-*d*:4,5-*d*’]bis([1,2,3]thiadiazole) (**19c**)

Violet solid, 36 mg (50%, procedure B) or 471 mg (57%, procedure D), eluent—CH_2_Cl_2_:hexane, 1:2 (*v*/*v*). R_f_ = 0.5 (CH_2_Cl_2_:hexane, 1:1 (*v*/*v*)). Mp > 250 °C. IR *ν*_max_ (KBr, cm^–1^): 1728, 1632, 1504, 1456, 1373, 1319, 1276, 1225, 1165, 1122, 1093, 1052, 824, 783, 701, 643, 548, 485. ^1^H NMR (300 MHz, CDCl_3_): *δ* 8.07 (d, *J* = 4.1 Hz, 2H), 7.39 (d, *J* = 4.1 Hz, 3H), 7.37 (d, *J* = 5.1 Hz, 2H), 7.12–7.09 (m, 3H). ^13^C NMR (75 MHz, CDCl_3_): *δ* 153.9, 142.7, 138.9, 136.4, 136.2, 131.9, 128.1, 125.8, 125.0, 124.7, 120.3. HRMS (ESI-TOF), *m*/*z*: calcd for C_22_H_11_N_4_S_6_ [M]^+^, 522.9302, found, 522.9299. MS (EI, 70 eV), *m*/*z* (*I*, %): 524 ([M + 2]^+^, 26), 523 ([M + 1]^+^, 25), 522 ([M]^+^, 100), 466 (90), 434 (35), 421 (32), 389 (30), 233 (80), 201 (50), 177 (45), 127 (30), 18 (60).

#### 3.12.12. 4,8-Bis(5-(2-ethylhexyl)thiophen-2-yl)benzo[1,2-*d*:4,5-*d*’]bis([1,2,3]thiadiazole) (**19d**)

Red solid, 55 mg (68%, procedure B) or 58 mg (72%, procedure D), eluent—CH_2_Cl_2_:hexane, 1:4 (*v*/*v*). R_f_ = 0.7 (CH_2_Cl_2_:hexane). Mp = 78–80 °C. IR *ν*_max_ (KBr, cm^–1^): 2957, 2924, 2855, 1664, 1376, 1313, 1272, 1245, 1139, 1088, 1013, 871, 816, 782, 703, 642, 545, 508. ^1^H NMR (300 MHz, CDCl_3_): *δ* 8.01 (d, *J* = 3.8, 2H), 7.00 (d, *J* = 3.8, 2H), 2.90 (d, *J* = 6.8, 4H), 1.73 (p, *J* = 5.9, 2H), 1.46–1.30 (m, 16H), 0.94–0.90 (m, 12H). ^13^C NMR (75 MHz, CDCl_3_): *δ* 153.7, 150.5, 138.6, 135.3, 131.1, 126.8, 120.3, 41.5, 34.4, 32.5, 28.9, 25.7, 23.0, 14.2, 10.9. HRMS (ESI-TOF), *m*/*z*: calcd for C_30_H_39_N_4_S_4_ [M + H]^+^, 583.2052, found, 583.2045. MS (EI, 70 eV), *m*/*z* (*I*, %): 584 ([M + 2]^+^, 4), 583 ([M + 1]^+^, 6), 582 ([M]^+^, 27), 525 (2), 497 (3), 483 (6), 427 (8), 343 (6), 328 (7), 252 (4), 177 (12), 164 (8), 121 (6), 57 (100), 41 (97), 29 (60).

#### 3.12.13. 4,8-Diphenylbenzo[1,2-*d*:4,5-*d*’]bis([1,2,3]thiadiazole) (**19e**)

Yellow solid, 24 mg (50%, procedure B) or 31 mg (65%, procedure D), eluent—CH_2_Cl_2_:hexane, 1:2 (*v*/*v*). R_f_ = 0.5 (CH_2_Cl_2_:hexane, 1:1 (*v*/*v*)). Mp > 250 °C. IR *ν*_max_ (KBr, cm^–1^): 1572, 1491, 1445, 1374, 1317, 1275, 1210, 1186, 1153, 1077, 1024, 978, 923, 898, 823, 766, 743, 696, 643, 539, 476. ^1^H NMR (300 MHz, CDCl_3_): *δ* 7 8.01 (d, *J* = 7.0, 4H), 7.69–7.58 (m, 6H). ^13^C NMR (75 MHz, CDCl_3_): *δ* 155.3, 141.7, 137.0, 130.0, 129.5, 129.2, 127.9. HRMS (ESI-TOF), *m*/*z*: calcd for C_18_H_11_N_4_S_2_ [M + H]^+^, 347.0420, found, 347.0421. MS (EI, 70 eV), *m*/*z* (*I*, %): 348 ([M + 2]^+^, 1), 347 ([M + 1]^+^, 2), 346 ([M]^+^, 7), 317 (3), 290 (100), 258 (10), 245 (4), 145 (30), 51 (3).

#### 3.12.14. 4,8-Di-*p*-tolylbenzo[1,2-*d*:4,5-*d*’]bis([1,2,3]thiadiazole) (**19f**)

Yellow solid, 28 mg (55%, procedure B) or 34 mg (65%, procedure D), eluent—CH_2_Cl_2_:hexane, 1:2 (*v*/*v*). R_f_ = 0.4 (CH_2_Cl_2_:hexane 1:1 (*v*/*v*)). Mp > 250 °C. IR *ν*_max_ (KBr, cm^–1^): 1607, 1446, 1373, 1311, 1277, 1209, 1182, 1119, 1101, 1021, 940, 898, 831, 802, 716, 652, 627, 582, 530, 491. ^1^H NMR (300 MHz, CDCl_3_): *δ* 7.90 (d, *J* = 7.9, 4H), 7.90 (d, *J* = 7.9, 4H), 2.52 (s, 6H). ^13^C NMR (75 MHz, CDCl_3_): *δ* 155.5, 141.7, 140.4, 134.3, 130.0, 129.6, 127.9, 21.6. HRMS (ESI-TOF), *m*/*z*: calcd for C_20_H_15_N_4_S_2_ [M + H]^+^, 375.0733, found, 375.0725. MS (EI, 70 eV), *m*/*z* (*I*, %): 374 ([M]^+^, 8), 318 (70), 303 (5), 285 (3), 159 (100), 115 (97), 93 (25), 63 (14), 39 (30).

#### 3.12.15. 4,8-Bis(4-methoxyphenyl)benzo[1,2-*d*:4,5-*d*’]bis([1,2,3]thiadiazole) (**19g**)

Orange solid, 32 mg (60%, procedure B) or 26 mg (50%, procedure D), eluent—CH_2_Cl_2_:hexane, 1:1 (*v*/*v*). R_f_ = 0.2 (CH_2_Cl_2_:hexane, 1:1 (*v*/*v*)). Mp > 250 °C. IR *ν*_max_ (KBr, cm^–1^): 1605, 1519, 1450, 1371, 1316, 1302, 1282, 1256, 1178, 1026, 816, 718, 546, 515. ^1^H NMR (300 MHz, CDCl_3_): *δ* 7.97 (d, *J* = 8.3, 4H), 7.17 (d, *J* = 8.3, 4H), 3.95 (s, 6H). HRMS (ESI-TOF), *m*/*z*: calcd for C_20_H_15_N_4_O_2_S_2_ [M + H]^+^, 407.0631, found, 407.0626. MS (EI, 70 eV), *m*/*z* (*I*, %): 407 ([M + 1]^+^, 1), 406 ([M]^+^, 10), 335 (100), 307 (25), 292 (12), 264 (8), 175 (20), 160 (21), 132 (98), 93 (6), 15 (70).

#### 3.12.16. 4,4′-(Benzo[1,2-*d*:4,5-*d*’]bis([1,2,3]thiadiazole)-4,8-diyl)bis(*N*,*N*-diphenylaniline) (**19h**)

Red solid, 63 mg (67%, procedure B) or 65 mg (69%, procedure D), eluent—CH_2_Cl_2_:hexane, 1:2 (*v*/*v*). R_f_ = 0.5 (CH_2_Cl_2_:hexane, 1:1 (*v*/*v*)). Mp > 250 °C. IR *ν*_max_ (KBr, cm^–1^): 1587, 1514, 1489, 1444, 1329, 1315, 1277, 1192, 1176, 841, 817, 753, 734, 695, 653, 620, 509. ^1^H NMR (300 MHz, CDCl_3_): *δ* 7.85 (d, *J* = 8.2, 4H), 7.35–7.30 (m, 8H), 7.24–7.08 (m, 16H). ^13^C NMR (75 MHz, CDCl_3_): *δ* 155.4, 149.4, 147.0, 141.2, 130.5, 129.6, 129.5, 127.0, 125.6, 124.0, 121.6. HRMS (ESI-TOF), *m*/*z*: calcd for C_42_H_29_N_6_S_2_ [M + H]^+^, 681.1890, found, 681.1901.

## 4. Conclusions

4,8-Dibromobenzo[1,2-*d*:4,5-*d*’]bis([1,2,3]thiadiazole) was successfully prepared in a moderate yield by the heating of a parent heterocycle with bromine in hydrobromic acid. Its structure was finally confirmed by single-crystal X-ray diffraction study. Aromatic nucleophilic substitution and palladium-catalyzed cross-coupling reactions were found to be powerful tools for the selective synthesis of various mono- and bis-derivatives. It was found that 4,8-dibromobenzo[1,2-*d*:4,5-*d*’]bis([1,2,3]thiadiazole) is resistant to the action of water, alcohols and corresponding alcoholates; when using aromatic nucleophilic substitution (S_N_Ar), mono- and bis-aminated derivatives were successfully obtained, while thiols formed only bis-derivatives, and the reaction could not be stopped at the stage of the formation of mono-thiols. The Stille coupling of 4,8-dibromobenzo[1,2-*d*:4,5-*d*’]bis([1,2,3]thiadiazole) is useful for the synthesis of bis-arylated heterocycles, while Suzuki–Miyaura coupling can be successfully employed for the selective formation of various mono- and di-(het)arylated derivatives of benzo[1,2-*d*:4,5-*d*’]bis([1,2,3]thiadiazole). The obtained compounds can represent an effective basis for the design and construction of components of organic light-emitting diodes and solar cells.

## Data Availability

Not applicable.

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
