# Peer review of "Efficient Synthesis of 4,8-Dibromo Derivative of Strong Electron-Deficient Benzo[1,2-d:4,5-d’]bis([1,2,3]thiadiazole) and Its SNAr and Cross-Coupling Reactions"

_molecules, 2022, doi:10.3390/molecules27217372_

Round 1

Reviewer 1 Report

The article by Rakitin O.A. et al. presents an in-depth study of the chemical properties of brominated benzo[1,2-d:4,5-d']bis([1,2,3]thiadiazoles).  The synthesis of the initial dibromo derivatives of bis-thiadiazoles and their further functionalization using nucleophilic substitution reactions of bromide anions as well as cross-coupling reactions are presented. Interesting products were obtained, which are of practical interest in the field of OLED technologies. The article is well written, all methods of physico-chemical analysis were used to characterize the substances, which confirms the authenticity of the structures. Based on this, I can recommend the article for publication in Molecules.

1. Figure 1 gives the HOMO-LUMO values, but does not specify the method of quantum calculation. It is necessary to add a comment.  

2. In the case of nucleophilic aromatic substitution in the dibrominated tricycle the monosubstitution proceeds better in comparison with complete substitution of halogen atoms. It is necessary to add a comment to the text about the reasons for this observation.

3. It is surprising that an effective electrophilic brominating agent such as NBS did not work in the tricycle bromination reaction. The authors do not explain the reason for this fact. Is it possible for the halogenation to proceed by the radical mechanism? Perhaps similar examples already exist in the literature for electron-deficient heteroaromatic compounds.

4. Some corrections may improve the quality of the article itself. For example, replace "diaryl(hetaryl)ated" with "di(het)arylated", "minimum amount" with "minor amounts", "electron-accepting" with "electron-deficient".

Author Response

Response to Reviewer 1.

The authors are grateful to the reviewer for a highly professional review.

Reviewer 1:

Figure 1 gives the HOMO-LUMO values, but does not specify the method of quantum calculation. It is necessary to add a comment.

Authors:

The HOMO-LUMO energies were calculated using the Gaussian 16 Rev C.01 program. M11 DFT functional with 6-31+g(d) basis. The information is added to the Experimental Section (Part 3.2.).

Reviewer 1:

In the case of nucleophilic aromatic substitution in the dibrominated tricycle the monosubstitution proceeds better in comparison with complete substitution of halogen atoms. It is necessary to add a comment to the text about the reasons for this observation.

Authors:

Nucleophilic aromatic substitution reactions under mild conditions lead to the formation of mono-substitution products. The introduction of a donor fragment into the dibromide molecule leads to a decrease in the reactivity of position 4 or 7 of the benzene ring. Therefore, to replace the second bromine atom, it is necessary to apply harsh conditions, for example, to heat the reaction mixture in DMF to 130 °C. The necessary comment has been added to the text of the paper.

Reviewer 1:

It is surprising that an effective electrophilic brominating agent such as NBS did not work in the tricycle bromination reaction. The authors do not explain the reason for this fact. Is it possible for the halogenation to proceed by the radical mechanism? Perhaps similar examples already exist in the literature for electron-deficient heteroaromatic compounds.

Authors:

Unfortunately, the reactivity of parent ultrahigh electron-withdrawing heterocycles, such as benzo-[1,2-c:4,5-c']bis[1,2,5]thiadiazole (benzo-bis-thiadiazole, BBT) or benzo[1,2-d:4,5-d']bis([1,2,3]thiadiazole) 2 has not been previously studied. We assume that for such compounds, the reactions of electrophilic substitution of the hydrogen atom are difficult and it is necessary to apply the conditions of radical reactions (for example, bromine in hydrobromic acid). The reactions of the unsubstituted tricycle will be considered in more detail in our next publication. The necessary comment has been added to the text of the paper.

Reviewer 1:

Some corrections may improve the quality of the article itself. For example, replace "diaryl(hetaryl)ated" with "di(het)arylated", "minimum amount" with "minor amounts", "electron-accepting" with "electron-deficient".

Authors:

Corrected as requested by the Reviewer.

Reviewer 2 Report

Manuscript Number: molecules-1999475

Full Title:  Efficient synthesis of 4,8-dibromo derivative of strong electron-accepting benzo[1,2-d:4,5-d']bis([1,2,3]thiadiazole) and its SNAr and cross-coupling reactions

Reviewer #1: The authors described the synthesis of hydrolytically and thermally stable 4,8-dibromobenzo[1,2-d:4,5-d']bis([1,2,3]thiadiazole) by bromination of its parent heterocycle and used as an intermediate for the synthesis of  mono- and diaryl(hetaryl)ated derivatives through Suzuki-Miyamura and stille cross-coupling reactions. Although the yield of 4,8-dibromobenzo[1,2-d:4,5-d']bis([1,2,3]thiadiazole) is moderate but it could be a useful intermediate in solar energy research. According to the spectroscopic data, all the compounds are very pure. However, there are some minor issues with this manuscript, as follows:

  • -(8-Bromobenzo[1,2-d:4,5-d']bis([1,2,3]thiadiazole)-4-yl)morpholine (12a): Rf = 0.1 (CH2Cl2:hexane, 335 1:1, (ν/ν)). ……(ν/ν) is looking like the Greek small letter nu (ν) not "v". Please check for such types of errors throughout the text.
  • 13C NMR (100 MHz, CDCl3): should be 13C NMR (75 MHz, CDCl3). Please check for such types of errors throughout the text.
  • Hz is missing in most of the J values. For example.:32 (dd, J = 7.5, 1.5, 2H). Please make the corrections throughout the text.
  • ,8-Bis(hexylthio)benzo[1,2-d:4,5-d']bis([1,2,3]thiadiazole) (15b) : 1.66 429 (p, J = 7.3, 4H)…what is p? Please check for such types of errors throughout the text.
  • Please check reference 38. …… Eur. J. Inorg. Chem. 2010, 2010, 4683–4696,
  • The HRMS spectra should be included in the supplementary file.

Author Response

Response to Reviewer 2.

The authors are grateful to the reviewer for a kind and highly professional review.

Reviewer 2:

-(8-Bromobenzo[1,2-d:4,5-d']bis([1,2,3]thiadiazole)-4-yl)morpholine (12a): Rf = 0.1 (CH2Cl2:hexane, 335 1:1, (ν/ν)). ……(ν/ν) is looking like the Greek small letter nu (ν) not "v". Please check for such types of errors throughout the text.

Authors:

We checked all the text and everywhere we used the Latin letters "v".

Reviewer 2:

13C NMR (100 MHz, CDCl3): should be 13C NMR (75 MHz, CDCl3). Please check for such types of errors throughout the text.

Authors:

Corrected as requested by the Reviewer.

Reviewer 2:

Hz is missing in most of the J values. For example.:32 (dd, J = 7.5, 1.5, 2H). Please make the corrections throughout the text.

Authors:

In the description of the general part of the experiment 3.2. we wrote that "J values are given in Hz". Therefore, we did not write the abbreviation Hz when describing each spectrum.

Reviewer 2:

,8-Bis(hexylthio)benzo[1,2-d:4,5-d']bis([1,2,3]thiadiazole) (15b) : 1.66 429 (p, J = 7.3, 4H)…what is p? Please check for such types of errors throughout the text.

Authors:

The MestReNova NMR program denoted by the letter p Quintet – five signals in the 1H NMR spectrum.

Reviewer 2:

Please check reference 38. …… Eur. J. Inorg. Chem. 2010, 2010, 4683–4696,

Authors:

Reference 38 replaced.

Reviewer 2:

The HRMS spectra should be included in the supplementary file.

Authors:

Added as requested by Reviewer.

Reviewer 3 Report

The authors elaborated the synthesis of a series of heterocyclic compounds. The compounds discussed in this paper shall be of interest to the researchers working in the area of heterocycles, OLEDs, etc.

I recommend the publication of this paper in MOLECULES.

Author Response

The authors are grateful to the reviewer for a kind and highly professional review.

There are no issues to reply.